# OpenReview forum: "Time-To-Inconsistency: A Survival Analysis of Large Language Model Robustness to Adversarial Attacks"
_ICLR.cc/2026/Conference — ICLR 2026 Poster_

### Official Review · Reviewer_YR9z · 2025-10-27

**Soundness:** 4
**Presentation:** 4
**Contribution:** 3
**Rating:** 8
**Confidence:** 2

**Summary:**

The paper models multi-turn LLM failures as a survival analysis modeling problem. With this, the authors demonstrate that the rate of failure within multi-turn LLM interactions are not constant hazard models, and that semantic drift is a strong predictor for failure.

**Strengths:**

I think the paper tackles an important consideration: as AI assistants are used more and more for long periods of time, how well do they remain consistent, especially under adversarial pressure. As far as I am aware this is high in originality.

The paper is clear, and well-written. The paper also is strong in general scholarship, suitably referencing existing literature appropriately, and nicely situates their work within the broader body of literature on AI evaluation, consistency, and survival analysis.

The experimental set-up is strong, with a range of metrics, survival models, and LLMs. The results shed light on the fact that the proportional hazards assumption doesn't necessarily hold across important features. This is a significant finding, particularly if further validated on additional datasets.

**Weaknesses:**

The limit of turn interactions to 8 steps is strangely limiting --- current models can engage with conversations over significantly larger context windows and more steps. If there are technical reasons for sticking to this -- e.g., if it were a facet of the dataset used or something similar, this should be explicitly mentioned in the paper.

The paper should explore a wider range of models. I think capturing a broader range of model sizes would be insightful because the survival times could be explored in the context of "scaling laws". I think the paper's analysis would likely explain newer models' ability to complete longer tasks (as identified by METR https://metr.org/blog/2025-03-19-measuring-ai-ability-to-complete-long-tasks/). Having a robust model explain this behaviour would be valuable.

The paper proposes that real-time monitoring of conversational drift could be employed to keep LLMs on track. This statement, while I think is likely, needs some empirical validation which the paper doesn't provide. That is, a demonstration of real-time monitoring would go a long way to validate the claim00

**Questions:**

Can you expand on and demonstrate the use of real-time monitoring for keeping LLMs consistent and on track?
What is the purpose of limiting the number of turn interactions to 8 steps?
How would you anticipate these time-to-inconsistent results scale with increased model size / parameters?

---

> ### Author Response · Authors · 2025-11-23
>
> Thank you for the thoughtful and positive assessment. We’re really glad you highlighted that **the paper tackles an important consideration** and that you found it clear and well-written. Even with the positive evaluation, we take all of your suggestions and concerns very seriously, and we were genuinely excited to share the results of the additional experiments you pointed to.
>
> ***
> We respond to the main weaknesses and questions below.
>
> >W1/Q1.2: The limit of turn interactions to 8 steps is strangely limiting --- current models can engage with conversations over significantly larger context windows and more steps. If there are technical reasons for sticking to this -- e.g., if it were a facet of the dataset used or something similar, this should be explicitly mentioned in the paper. What is the purpose of limiting the number of turn interactions to 8 steps?
>
> We agree that modern LLMs can in principle sustain much longer conversations, and here's the reason with select such setting:
>
> - Dataset design and effective sample size. MT-Consistency is constructed with a maximum of 8 adversarial follow-ups per base question and per model. After following the benchmark protocol (700 questions × 9 LLMs × 8 follow-ups) and restricting to conversations where the initial answer is correct (round 0), we still obtain **36,951 turns** QA content. This gives us a substantial and statistically meaningful sample for estimating hazards and drift effects.
>
> - Alignment with adversarial protocols. The choice of 8 turns is not arbitrary on our side: MT-Consistency defines 8 distinct adversarial prompt types (closed-ended, misleading, emotional appeal, expert appeal, etc.), many of which are adapted from prior work in LLM robustness and psychology. These are explicitly designed to stress-test confidence and stability over a short but structured adversarial sequence, which aligns well with our goal of modeling time-to-inconsistency under targeted pressure. For that reason, we followed the original benchmark design rather than redefining the horizon.
>
> We fully agree that longer, mixed-initiative dialogues are important. In the revised paper, we now call out the 8-turn limit explicitly in the Limitations section and discuss extending time-to-inconsistency analysis to longer conversations and other datasets as a key direction for future work.
>
> >W2: The paper should explore a wider range of models. I think capturing a broader range of model sizes would be insightful because the survival times could be explored in the context of "scaling laws". I think the paper's analysis would likely explain newer models' ability to complete longer tasks (as identified by METR https://metr.org/blog/2025-03-19-measuring-ai-ability-to-complete-long-tasks/). Having a robust model explain this behaviour would be valuable.
>
> Thank you for this suggestion—we genuinely like the idea of connecting time-to-inconsistency with scaling-law style behavior. In this paper we chose to work with the 9 models already included in MT-Consistency, which span several major families and capability levels, but we agree that explicitly varying model size and “reasoning depth” would add an important extra dimension!!
>
> Due to compute and time constraints, a full scaling-law study is beyond the scope of this initial work, but we completely agree with your intuition and would love to mark it as a key following step. We see our paper as a first demonstration that survival analysis gives meaningful, temporally-resolved signals for multi-turn robustness; extending the same framework to a size-graded family of models is exactly the kind of follow-up we hope to pursue, ideally in collaboration with others in the community.

---

> ### Author Response · Authors · 2025-11-23
>
> >W3/Q1.1: The paper proposes that real-time monitoring of conversational drift could be employed to keep LLMs on track. This statement, while I think is likely, needs some empirical validation which the paper doesn't provide. That is, a demonstration of real-time monitoring would go a long way to validate the claim. Can you expand on and demonstrate the use of real-time monitoring for keeping LLMs consistent and on track?
>
> Thank you for pointing this out — we agree that our claim about using drift monitoring as a real-time safeguard should be backed by an explicit empirical demonstration. Here we go! We are excited to share the results of such experiment.
>
> **Action Taken**: In response to your request, we have added a new experimental section (Section 5.4) and Table 4 to the revised manuscript. We implemented a retrospective risk monitor using our trained Weibull AFT model to calculate a Conditional Failure Probability (CFP) at every turn.
>
> Results: The results (detailed as below) strongly validate the feasibility of real-time monitoring:
>
> | Group        | Method           | % alerted | Alerts / conv. | First-alert round | Failure round |
> |--------------|------------------|----------:|----------------:|------------------:|--------------:|
> | All (140)    | AFT (ours)       |     55%   |            1.1  |               4.0 |      --       |
> |              | Drift baseline   |     51%   |            1.3  |               4.0 |      --       |
> | Failing (88) | AFT (ours)       |     76%   |            1.4  |               3.3 |       5.7     |
> |              | Drift baseline   |     62%   |            1.6  |               3.9 |       5.7     |
> | Censored (52)| AFT (ours)       |     19%   |            0.5  |               5.2 |      --       |
> |              | Drift baseline   |     32%   |            1.2  |               4.2 |      --       |
>
> - High Sensitivity: The AFT monitor successfully triggers an alert for 76% of failing conversations before the first inconsistent answer occurs.
>
> - Actionable Lead Time: Crucially, for the flagged conversations, the monitor provides a median lead time of 2 turns (mean 2.3) between the first alert and the failure. This window is sufficiently wide to trigger interventions (e.g., activating a verifier, asking clarifying questions, or resetting context) to "keep the LLM on track."
>
> - Low False Positive Rate: The system is operationally selective. It alerts on only 19% of safe (censored) conversations.
>
> - Superiority over Baselines: We compared this against a "Drift-Threshold Baseline" (alerting solely on raw semantic drift). The AFT model detects significantly more failures (76% vs. 62%) while generating far less noise on safe conversations (0.5 alerts/conv vs. 1.2 alerts/conv).
>
> These findings empirically demonstrate that the accelerating hazard modeled by our framework translates into a concrete, early-warning signal that can be used to safeguard LLM consistency in real-time applications.
>
>
>
> >Q1.3 How would you anticipate these time-to-inconsistent results scale with increased model size / parameters?
>
> We appreciate this forward-looking question. Conceptually, our framework makes a clear prediction: if larger models better maintain semantic coherence and resist adversarial drift, their hazard curves should shift downward and their survival functions should decay more slowly, leading to a longer expected time-to-inconsistency. In other words, under a fixed attack protocol and comparable alignment, we would expect larger models to (i) fail less often within an 8-turn horizon and (ii) exhibit later failure times when they do fail. At the same time, we do not expect this relationship to be purely monotone in practice, because real systems differ not only in parameter count but also in alignment training and safety policies; strongly instruction-following models may be more susceptible to certain persona-based attacks even when they are more capable overall.
>
> **Again, we sincerely appreciate your positive assessment of our work—it means a great deal to us, and we remain committed to further advancing the robustness of modern AI/LLM systems.**

---

> > ### Comment · Reviewer_YR9z · 2025-11-26
> >
> > Thank you for clarifying the reason for limiting the interaction turns to 8 steps. That clarifies things for me there.
> > The new experiments also fill me with more confidence that my initial assessment was correct. I will be maintaining my original score.

---

> ### Author Response · Authors · 2025-11-26
>
> Thank you again, Reviewer YR9z, for your thoughtful engagement with our work and for confirming that the new experiments strengthened your confidence in your original assessment. We’re glad that the clarification on the 8-turn limit and the real-time monitoring results addressed your concerns.
>
> If you have any further questions or would like more detail on any part of the work, we’d be very happy to discuss within the review process. If the new results have increased your confidence, we would highly appreciate your **consideration of an updated confidence score**, as it would be very meaningful to us as well.
>
> Happy Thanksgiving, and thank you again for your careful review!

---

### Official Review · Reviewer_y2g9 · 2025-10-29

**Soundness:** 3
**Presentation:** 1
**Contribution:** 2
**Rating:** 2
**Confidence:** 2

**Summary:**

This paper applies survival analysis to multi-turn LLM conversations, measuring the time (in turns) until the LLM produces a wrong answer (failure).

**Method**: The authors first construct a feature vector for every turn in a multi-turn conversation, and then fit three classes of survival analysis models to those features (two Cox proportional hazards (PH) models, four accelerated failure time (AFT) models, and a random survival forest). Failure is defined as the LLM giving an incorrect answer.

Using sentence-transformer embeddings of prompts, the features for each conversation turn are
1. prompt-to-prompt drift: cosine dissimilarity between the current and previous prompt embedding
2. context-to-prompt drift: cosine dissimilarity between the current and the (mean) previous prompt embeddings
3. cumulative drift: the sum of prompt-to-prompt drifts up to the current turn
4. discrete covariates: domain of the conversation topic, the question difficulty, and the model/model family
To the best of my understanding, this only considers *user* prompts; the features do *not* consider LLM responses.

**Experiments**: The authors apply all methods to the MT-Consistency benchmark, which contains multi-turn conversations for 9 LLMs. The experiments only use conversations with an initially correct answer (hence measuring if the LLM can be "swayed") and split those conversations into a 80%-20% train-test split.

The paper then evaluates how well different survival analysis models predict failure. AFT models are overall more accurate and better calibrated, likely because the proportional hazards assumption of PH models is violated for crucial features. The authors use the proportional hazards model to analyze their features; they find that drift between subsequent conversation turns increase the risk most, while the cumulative drift over a conversation (e.g., a gradual topic change) reduces the risk (in that PH model).

**Takeaways**: An abrupt change in semantics (in terms of prompt embedding similarity) increases the risk of LLMs providing a wrong answer, while a gradual change over many terms can actually reduce the risk. In particular, this implies that changes in topic are not detrimental (but can be good) as long as they are gradual. The risk of failure is not constant over turns, but changes as a conversation progresses. Lastly, the authors propose to use a lightweight AFT models in production to detect and intervene prompts where the risk of failure dramatically increases.

**Strengths:**

1. The results hint that lightweight AFT models could be a simple additional safeguard for practical LLM deployments. I'm curious to see if their efficacy holds up in an experiment.
2. The author's core motivation, moving away from single-turn evaluations or static metrics for multi-turn conversations, is clear and reasonable. The related work section provides a good contextualization of this motivation.
3. Similarly, the general idea of evaluating multi-turn LLM conversations from the perspective of survival analysis is novel and interesting.
4. The method description is very self-contained and detailed. Thus, the paper is also accessible to readers that might be less familiar with survival analysis.

**Weaknesses:**

While I'm not an expert in survival analysis, I could understand the overall methods in the paper. However, the paper could greatly benefit from a systematic overhaul of the presentation; I found it challenging to understand what the actual goals and contributions are, and I am still unsure about key parts of the feature engineering. In addition, the paper's takeaways are relatively narrow and could benefit from more discussion or experiments.

**Paper presentation**: The paper's overall goal/contribution could be presented more clearly. The abstract and introduction led me to believe that the paper's primary aim is to *analyze LLMs in multi-turn conversations* using survival analysis (e.g., L17-18, L58-59). However, the bulk of the paper focuses on *designing survival analysis models*, and the experiments evaluate *those methods* (not LLMs). In fact, only the last two pages contain any results for LLMs, without focusing on individual architectures or families, and the main takeaways and contributions became only clear to me when reading the discussion. Crucially, the key takeaway that survival analysis could be used as a practical safeguard, is never mentioned until the last paragraph of the discussion. The paper would greatly benefit from mentioning the actual goal and contributions early, so that readers understand how to interpret the method and results.

**Confusion around feature engineering**: Potentially due to the presentation issues, I found the first three features in Section 3.2 (Equations 1-3) difficult to understand. In particular, the writing should clarify whether drift uses the embeddings of *user prompts*, *LLM completions*, or a *concatenation* of the two. I.e., the paper should make it clear whether survival analysis is done w.r.t. to user inputs or LLM outputs; this is never mentioned explicitly. Adding to this confusion is the phrasing in Section 3.2: $e_t$ is multiple times defined as the *user* prompt embedding (L146, L152), but Prompt-to-Prompt drift mentions the shift between *conversational turns* (L149-150), which could entail a user prompt followed by an LLM response or vice versa. Similarly, $\mathbf{\bar{e}_{1:t-1}}$ is both referenced as the mean embedding of all previous *prompts* (L147-148), but then later seems to entail the full conversation context (which I initially presumed to include the LLM's responses) on L157.
Clarity about what the features entail is very important in my opinion, as all of the paper's insights relate around the semantic dissimilarity of user/LLM prompts. In my initial read, I understood the features to be about the concatenation of user prompts and LLM responses; after studying the paper more, I now believe features only consider user prompts, but I am still unsure.

**Significance**: The paper's contributions in their current form are a bit limited and could be expanded through more discussion or experiments. First, while having evidence for it is nice, the insight that an abrupt change in the conversation is more likely to induce LLM failure is in itself not surprising. Second, the finding that a gradual change in semantics might help robustness is surprising, but could benefit from more depth and additional evidence beyond the hazard ratio of a PH model (e.g., a controlled study). Lastly, the main practical takeaway in my opinion (and ultimately the reason to apply survival analysis in the first place) is that AFT models could serve as safeguards. However, this is only mentioned in one paragraph (L464-471); I would find additional experiments and depth on this very exciting.

**Questions:**

1. Are the sentence-transformer embeddings in Section 3.2 only taken over user prompts? Or are they calculated differently?
2. Section 5.4 and Figure 1 show that Prompt-to-Prompt drift is an important feature for failure prediction. Similarly, the fact that cumulative drifts are protective (one of the main insights of this paper) seem to stem from the same analysis. However, this is done w.r.t. a PH model, and the PH assumption seems not to hold for Prompt-to-Prompt drift (Table 2). How accurate are the insights given the assumption mismatch?
3. The results in Section 5.1/Tables 1 and 2 mention model-drift interaction terms for AFT models, but Section 3.3 only mentions such a term for the PH models. What is the interaction term for AFT models?

---

> ### Author Response · Authors · 2025-11-23
>
> We thank the reviewer for the clear summary of our work and for the thoughtful comments on our methodology, experiments, and main takeaways. We are especially glad to see your interest in whether lightweight AFT models hold up in practice, and we appreciate your positive remarks on the core motivation of this work, related work section and the level of detail in the methods—we indeed aimed to make the paper accessible to readers with varying backgrounds in survival analysis. We are also grateful for the weaknesses and concerns you raised, particularly regarding presentation, clarity of our feature engineering, and the need to better highlight the significance and practical impact of our findings. Below, we respond to each of these points in detail.
>
> ***
> > W1: Paper presentation: The paper's overall goal/contribution could be presented more clearly. The abstract and introduction led me to believe that the paper's primary aim is to analyze LLMs in multi-turn conversations using survival analysis (e.g., L17-18, L58-59). However, the bulk of the paper focuses on designing survival analysis models, and the experiments evaluate those methods (not LLMs). In fact, only the last two pages contain any results for LLMs, without focusing on individual architectures or families, and the main takeaways and contributions became only clear to me when reading the discussion. Crucially, the key takeaway that survival analysis could be used as a practical safeguard, is never mentioned until the last paragraph of the discussion. The paper would greatly benefit from mentioning the actual goal and contributions early, so that readers understand how to interpret the method and results.
>
> Thank you for pointing this out! We agree that the original version did not foreground the main goals and takeaways clearly enough. In the revision, we explicitly restructured the Abstract and Introduction to:
> - Clearly state the primary goal: to understand time-to-inconsistency of LLMs in adversarial multi-turn conversations and to show how survival analysis yields operational risk signals, not just a new loss function.
> - Make the three main contributions explicit and early (Introduction, end of Sec. 1):
>   - Formalizing multi-turn robustness as a survival/time-to-event problem.
>   - Showing drift-aware dynamics (abrupt vs. cumulative shift) and their impact on hazard.
>   - Demonstrating that lightweight AFT models can function as practical, turn-level risk monitors.
>
> We also reorganized the narrative so that the work is clearly about LLM behavior over time, rather than “yet another comparison of survival models.” The survival models are presented as tools to study LLM failure trajectories; we now **emphasize LLM-side insights and safeguards from the Introduction to the Results and Discussion sections**, not only at the very end.
>
> Finally, we added a subsection showing experiment results on AFT application, Sec. 4.5 “Retrospective Risk Monitoring with AFT”, that explicitly develops and evaluates the “AFT as safeguard” angle, rather than mentioning it only in a closing paragraph.
>
> >W2/Q1: "Confusion around feature engineering: I found the first three features in Section 3.2 (Equations 1-3) difficult to understand. In particular, the writing should clarify whether drift uses the embeddings of user prompts, LLM completions, or a concatenation of the two. I.e., the paper should make it clear whether survival analysis is done w.r.t. to user inputs or LLM outputs; this is never mentioned explicitly..."
>
> We very much appreciate this comment—the original wording was indeed too easy to misread. To response concisely, P2P using user prompt embeddings, and content is gained by take both user prompt and model response into consideration. And the C2P is calculated based on the embedding of all existing (user prompts+model response)x(t-1) and current round user prompt. And cum is the sum of P2P.
> To make it crystal clear, in the revised Sec. 3.2 (Predictive Feature Engineering), we now clearly distinguish:
> - Prompt embeddings: embedding of the user prompt at turn 𝑡.
> - Context embeddings: context(i,t) is the full dialogue prefix up to and including turn t.
>
> Drift features are then:
> - P2P: user–user change between consecutive prompts.
> - C2P: misalignment between the full conversational context till last round and the current user prompt.
> - Cum: the total user-side semantic distance traveled
>
> Formula details are all updated in the revised version - Sec. 3.2

---

> ### Author Response · Authors · 2025-11-23
>
> >W3-1: While having evidence for it is nice, the insight that an abrupt change in the conversation is more likely to induce LLM failure is in itself not surprising
>
> We agree that the idea that abrupt shifts can be harmful is intuitively plausible. Our contribution here is to (i) turn this intuition into quantitative, model-comparative evidence across 9 LLMs and 8 adversarial strategies, and (ii) show how large the effect is relative to other covariates. In the revised manuscript (Sec. 4.3), we now show that prompt-to-prompt drift is the dominant driver of hazard, with hazard ratios up to 4–5× for some models, even after controlling for subject domain, difficulty, context-to-prompt drift, and cumulative drift. We also demonstrate that this pattern is consistent across Cox, AFT, and RSF (variable-importance) views. To our knowledge, prior work has not provided such a temporally resolved, cross-model quantification of how much acute drift changes failure risk, nor how its effect compares to other conversational signals.
>
> >W3-2: The finding that a gradual change in semantics might help robustness is surprising, but could benefit from more depth and additional evidence beyond the hazard ratio of a PH model (e.g., a controlled study)
>
> We agree that this surprising result merits more depth. In the revision we go beyond a single PH hazard ratio and provide converging evidence from multiple angles (Sec. 4.3–4.4):
> - The protective effect of cumulative drift (HR < 1) appears consistently in both Cox and AFT models, and is echoed by RSF variable importance.
> - We stratify conversations by risk/drift levels and compare survival curves (Table 3), showing that high-drift trajectories that survive early turns indeed exhibit longer median survival and lower subsequent hazard.
>
> Together, these analyses strengthen the claim that cumulative drift is not an artifact of one model family. A fully controlled synthetic study (e.g., constructing matched conversations with only drift patterns varied) is an exciting direction, but beyond the scope of this first large-scale survival analysis. We will definitely combine it into our research in the following steps.
>
> >W3-3: the main practical takeaway in my opinion (and ultimately the reason to apply survival analysis in the first place) is that AFT models could serve as safeguards. However, this is only mentioned in one paragraph (L464-471); I would find additional experiments and depth on this very exciting.
>
> As responsed for the paper presentation, we've highlighted such findings explicitly in many sections of the revised version. Plus, an additional experiment was conducted and the results and discussion have been updated to the new subsection 5.5 RETROSPECTIVE RISK MONITORING WITH AFT. Feel free to check the revised paper for full details. For your convenience, here's the results summary and the main observations:
> The results (detailed as below) strongly validate the feasibility of real-time monitoring:
>
> | Group        | Method           | % alerted | Alerts / conv. | First-alert round | Failure round |
> |--------------|------------------|----------:|----------------:|------------------:|--------------:|
> | All (140)    | AFT (ours)       |     55%   |            1.1  |               4.0 |      --       |
> |              | Drift baseline   |     51%   |            1.3  |               4.0 |      --       |
> | Failing (88) | AFT (ours)       |     76%   |            1.4  |               3.3 |       5.7     |
> |              | Drift baseline   |     62%   |            1.6  |               3.9 |       5.7     |
> | Censored (52)| AFT (ours)       |     19%   |            0.5  |               5.2 |      --       |
> |              | Drift baseline   |     32%   |            1.2  |               4.2 |      --       |
>
> - High Sensitivity: The AFT monitor successfully triggers an alert for 76% of failing conversations before the first inconsistent answer occurs.
>
> - Actionable Lead Time: Crucially, for the flagged conversations, the monitor provides a median lead time of 2 turns (mean 2.3) between the first alert and the failure. This window is sufficiently wide to trigger interventions (e.g., activating a verifier, asking clarifying questions, or resetting context) to "keep the LLM on track."
>
> - Low False Positive Rate: The system is operationally selective. It alerts on only 19% of safe (censored) conversations.
>
> - Superiority over Baselines: We compared this against a "Drift-Threshold Baseline" (alerting solely on raw semantic drift). The AFT model detects significantly more failures (76% vs. 62%) while generating far less noise on safe conversations (0.5 alerts/conv vs. 1.2 alerts/conv).

---

> ### Author Response · Authors · 2025-11-23
>
> >Q2: Section 5.4 and Figure 1 show that Prompt-to-Prompt drift is an important feature for failure prediction. Similarly, the fact that cumulative drifts are protective (one of the main insights of this paper) seem to stem from the same analysis. However, this is done w.r.t. a PH model, and the PH assumption seems not to hold for Prompt-to-Prompt drift (Table 2). How accurate are the insights given the assumption mismatch?
>
> We thank the reviewer for this rigorous statistical observation. You are correct that the Proportional Hazards (PH) assumption is violated for Prompt-to-Prompt (p2p) drift, which could theoretically bias the hazard ratios in a standard Cox model.To address this and verify the accuracy of our insights, we performed a robustness check using a Log-Normal Accelerated Failure Time (AFT) model, which does not rely on the PH assumption. We compared the risk profiles generated by both models in the updated Figure 1 (Section 5.4).
>
> The results demonstrate strong concordance between the two frameworks, confirming that our qualitative insights are robust:
> - Prompt-to-Prompt (p2p) is a confirmed risk: While the Cox model assigns a high Hazard Ratio ($HR > 1$), the AFT model assigns a low Acceleration Factor ($AF < 1$). Since $AF < 1$ implies "time accelerates" (i.e., survival time shortens), both models agree that acute p2p shifts significantly increase failure risk.
> - Cumulative Drift is confirmed protective: The Cox model shows a protective effect ($HR < 1$), which is corroborated by the AFT model showing time expansion ($AF > 1$). For example, in Gemini, Cox estimates $HR \approx 0.2$ while AFT estimates $AF \approx 2.1$, both indicating increased resilience.
>
> We have updated Section 5.4 to include this AFT analysis, which confirms that the reported drift–hazard relationships are genuine phenomena and not artifacts of model misspecification.
>
> >Q3: The results in Section 5.1/Tables 1 and 2 mention model-drift interaction terms for AFT models, but Section 3.3 only mentions such a term for the PH models. What is the interaction term for AFT models?
>
> Thank you for pointing out this inconsistency in the description. Our AFT models use the **same interaction structure** as the advanced Cox PH models; the only change is the link function (log-hazard vs. log-time).
>
> We have updated Section 3.3 to state explicitly that both the Cox PH and the AFT specifications use this interaction design; they differ only in the assumed error distribution and interpretation (hazard ratios vs. acceleration factors). The table 1 also shows that # of covariates for Cox/AFT model with interactions are equal.(53)
>
> **We appreciate your thoughtful feedback in strengthening our work. We hope the carefully revised manuscript and responses address your concerns sufficiently, and we respectfully ask that you consider raising your rating in light of these improvements.**

---

> ### Comment · Reviewer_y2g9 · 2025-11-24
> **Response to rebuttal**
>
> I thank the authors for their detailed reply. I checked the updated version of the paper, the other reviews, and other rebuttals.
>
> **Re W1**: I think the presentation is now clearer than before, and the structure of the paper is more coherent. The introduction now also makes it clear that this is *not* a benchmark of LLMs but about survival analysis for LLMs.
>
> **Re W2/Q1**: I appreciate the overhaul of Sec. 3.2; I can now understand what the features actually entail. I think all the previous inconsistencies are now resolved.
>
> **Re W3-1 and W3-2**: I thank the authors for the clarifications and agree that providing quantitative evidence for the idea that abrupt shifts can be harmful is useful. However, my main issue is that all explanations are model-based, whereas results with a causal intervention would be much stronger. Of course, I appreciate the authors' transparency that this is infeasible in the context of the remaining reviewing period, and do not expect them to actually perform such experiments right now. Nevertheless, this remains a non-trivial limitation of the work.
>
> **Re W3-3**: I appreciate the retroactive study of monitoring, and I agree that this adds substance to the paper's claims. However, as in my previous point, I believe that a more broad study (which is out-of-scope for the current submission) would strengthen the contribution a lot. Additionally, at a "true positive rate" of 76% and false positive rate of 19%, it is unclear to me whether an AFT-based detector would ultimately be practically feasible (even with much additional engineering effort).
>
> **Re Q2 and Q3**: I thank the authors for their explanations, which resolve those questions.
>
> I raised my scores given the significant improvements in presentation and the clarifications on the contribution. However, I still believe that the current state of this work falls short in its contribution and requires work beyond the current reviewing cycle.

---

> ### Author Response · Authors · 2025-11-24
>
> We appreciate your recognition of the improved presentation and clarified contributions, and for revisiting and raising your overall assessment. We fully acknowledge your points regarding causal interventions and broader monitoring studies—these are valuable directions we're committed to pursuing in future work.
>
> We're grateful for your engagement and the thoughtful guidance that strengthened this work!

---

### Official Review · Reviewer_KPhd · 2025-10-30

**Soundness:** 3
**Presentation:** 2
**Contribution:** 3
**Rating:** 6
**Confidence:** 2

**Summary:**

This paper applies survival analysis to model when and why LLMs fail during multi-turn conversations. The key finding: abrupt prompt-to-prompt semantic shifts increase failure risk by 2-4.6, while gradual cumulative drift is protective (reduces risk). This means the velocity of conversational change matters more than total distance traveled—models can adapt to gradual topic evolution but break under sudden semantic jumps. Accelerated Failure Time models achieve 87.4% accuracy in predicting failures with 48% better calibration than baselines, enabling real-time monitoring systems that detect dangerous conversational shocks before failure occurs.

**Strengths:**

Reframes multi-turn robustness as a time-to-event problem and is (to my knowledge) the first comprehensive use of survival analysis (Cox/AFT/RSF) for LLM dialogue failure.

Introduces semantic drift features (prompt-to-prompt, context-to-prompt, cumulative) as time-varying covariates—an inventive, interpretable way to capture conversational dynamics.

Rigorous 36,951-turn analysis across 9 models. Multi-paradigm approach (Cox, AFT, RSF) ensures robust conclusions.

Establishes survival analysis as a powerful paradigm for evaluating conversational robustness, likely to influence future benchmarks and tooling.

Produces actionable design guidance (detect/mitigate abrupt drift; use AFT-style monitors for real-time risk stratification).

**Weaknesses:**

Entire analysis uses MT-Consistency benchmark with one adversarial protocol.

Paper doesn't distinguish why models fail—sycophancy vs. context confusion vs. knowledge gaps all labeled as "failure."

**Questions:**

Your correlation between abrupt P2P drift and failure is compelling, but could high P2P simply indicate harder follow-up questions rather than causing failure?

Why does cumulative drift protect models?

---

> ### Author Response · Authors · 2025-11-23
>
> Thank you for the thoughtful review and for summarizing the main ideas of the paper so clearly. We’re glad the overall framing and semantic-drift features came through, and we’ll try to address your concerns point by point.
> ***
> > W1:Entire analysis uses MT-Consistency benchmark with one adversarial protocol.
>
> Thank you for pointing this out. You’re right that all of our empirical results come from MT-Consistency, and we now say this much more explicitly in the revised Data and Limitations sections. Our goal in this first study was to keep the setting controlled: MT-Consistency gives a fixed 8-turn adversarial script with 700 questions across 39 subjects and eight designed attack patterns (Closed, Open, Misleading, Emotional Appeal, Impolite Tone, Expert Appeal, Consensus Appeal, False Agreement).
>
> At the same time, we want to emphasize that—even though the benchmark is “one dataset”—the induced interaction log is actually large: once we apply the MT-Consistency filtering (only conversations that start correct) across 9 models and 8 adversarial turns, we end up with **36,951** turns of evaluated interaction. This gives us **enough statistical power** to fit and compare Cox, AFT, and RSF models, and to stratify by model and risk level in a meaningful way.
>
> We fully agree, however, that using a single benchmark and protocol limits how far we can push generalization. In the revision, we explicitly call this out as a main limitation in the Discussion and reframe our contribution as a conceptual and methodological step: showing that survival analysis is a useful lens on multi-turn robustness and that simple drift features act as strong time-varying covariates in at least one realistic adversarial setting. We also added a short future-work paragraph outlining our plan to plug the same pipeline into other multi-turn benchmarks and longer, less scripted conversations, to test how stable these patterns are beyond MT-Consistency.
>
>
> > W2: “Paper doesn't distinguish why models fail—sycophancy vs. context confusion vs. knowledge gaps all labeled as 'failure'.”
>
> Our primary goal here was to see whether the survival-analysis lens itself is useful for multi-turn robustness: can we treat “does the conversation stay correct?” as a time-to-event process, plug in simple drift features as time-varying covariates, and get a hazard signal that is strong enough to be useful for robustness monitoring? For that first step, we stayed aligned with the MT-Consistency benchmark and treated “first inconsistent answer under their scoring” as a single event, regardless of whether it came from sycophancy, confusion about context, or a knowledge gap. This keeps the event definition clean and matches the intuition that, from a safety/trust perspective, the occurrence of the first failure already matters a lot, no matter which mechanism caused it.
>
> At the same time, we cannot agree with you more on the in-depth analysis on why models fail - being inconsistent under our settings. While, the original data was mainly collected based on large language model instead of large reasoning models. Many of times, the LLMs sway without explicitly state the reasons or reasoning, which set obstacle on exploring the root cause. We are actively running the whole experiment on the large reasoning model set so that more reasoning could be captured and analyzed. We see this as one critical step for the development of LLM/LRM robustness.

---

> ### Author Response · Authors · 2025-11-23
>
> >W3: could high P2P simply indicate harder follow-up questions rather than causing failure?
> Thank you for raising this — we agree this is an important confound to think about.
>
> In our setup, high P2P drift is defined purely as the cosine distance between the embeddings of two consecutive prompts, i.e., how much the semantic content of the user’s request jumps between turns, not how “hard” the question is per se. Question difficulty is modeled separately. Concretely, the MT-Consistency difficulty band (Elementary / High School / College / Professional) enters the survival models as its own set of categorical covariates, and the hazard ratios we report for P2P drift are conditional on those difficulty indicators (as well as subject domain and model identity).
>
> Empirically, we include **difficulty as a covariate**. While its effect on the hazard is **much smaller** than the effect of P2P drift: the large hazard ratios we report for P2P are not explained away by the difficulty band. Intuitively, this matches what we see in the benchmark: even within the same difficulty level, conversations that make sudden topical jumps are much more likely to fail than those that move gradually.
>
>
> > W4: Why does cumulative drift protect models?
>
> We agree this is the most counterintuitive (and honestly, to us, the most interesting) result in the paper. Our current view is that the “protective” effect of cumulative drift is fundamentally a conditional one. In a survival model, conversations with high cumulative drift at turn 𝑡 are exactly those that have already wandered quite far without failing yet. So when we estimate the effect of cumulative drift on the hazard, we are conditioning on survival up to 𝑡: we are looking at a subset of dialogues where the model has successfully adapted through several shifts. In that subset, additional drift tends to be less dangerous, because the model is already operating in the “new” semantic region and abrupt shocks become less surprising. Put differently, prompt-to-prompt drift acts like an acute shock that spikes short-term risk, while cumulative drift behaves more like a resilience marker: given that a conversation has already absorbed a lot of drift and stayed consistent, the incremental hazard is lower. We see it as an invitation for follow-up work—e.g., from domain adaptation or information-theoretic perspectives—to develop deeper explanations of why some conversations become more robust as they evolve.
>
> **We appreciate your thoughtful feedback in strengthening our work. We hope the carefully revised manuscript and responses address your concerns sufficiently, and we respectfully ask that you consider raising your rating in light of these improvements.**

---

> > ### Author Response · Authors · 2025-11-26
> >
> > Dear Reviewer KPhd,
> >
> > Thank you for your constructive review and for accurately summarizing our contributions regarding the application of survival analysis to LLM robustness.
> >
> > As the discussion period ends, we would like to provide a concise summary of how we addressed your specific questions and concerns in the revision:
> >
> > **W1 (Single Benchmark)**: We acknowledged this limitation in the Discussion. However, we clarified that while the base questions are fixed, the interaction with 9 models over 8 adversarial turns generates ~37k data points. This provides sufficient statistical power to validate the survival analysis methodology, which can now be applied to other benchmarks in future work.
> >
> > **W2 (Failure Types)**: We explained that we prioritize the "first failure" event because, in safety-critical contexts, the loss of trust occurs regardless of the root cause (sycophancy vs. knowledge gap). We also noted that we are currently extending this work to Large Reasoning Models (LRMs) to better isolate reasoning traces for root-cause analysis.
> >
> > **Q1 (P2P Drift vs. Difficulty)**: We clarified that question difficulty (e.g., Elementary vs. Professional) is explicitly included as a covariate in our model. The high hazard ratio for Prompt-to-Prompt (P2P) drift exists conditional on difficulty—meaning sudden semantic jumps increase risk even when the question difficulty remains constant.
> >
> > **Q2 (Protective Cumulative Drift)**: We offered an interpretation of this counterintuitive finding: Cumulative drift acts as a "resilience marker." Conversations with high cumulative drift are those that have already successfully adapted to new semantic regions. P2P drift acts as an acute shock, whereas cumulative drift represents proven adaptability.
> >
> > We hope these clarifications regarding the variable controls and the "shock vs. resilience" dynamic address your concerns. We respectfully ask that you **consider confirming or raising your score based on these revisions/what we discussed**.

---

### Official Review · Reviewer_t9mq · 2025-10-31

**Soundness:** 2
**Presentation:** 1
**Contribution:** 2
**Rating:** 4
**Confidence:** 3

**Summary:**

This work presents the first large-scale survival analysis of conversational AI robustness, modeling failure as a time-to-event process across 36,951 conversation turns and nine LLMs.

The study reveals that abrupt semantic drift sharply increases failure risk, whereas gradual, cumulative drift helps maintain longer, more stable dialogues.

These findings establish survival analysis as a powerful new framework for evaluating LLM robustness and guiding the design of resilient conversational agents.

**Strengths:**

1. This paper proposes a multi-turn assessments for LLM robustness.

**Weaknesses:**

1. ***Missing Conceptual Depth***
- The introduction states the idea of using survival analysis but doesn’t explain why this framework is particularly appropriate or transformative for conversational robustness.

- It should elaborate how “time-to-failure” parallels conversational degradation (e.g., semantic drift, confidence decay, compounding reasoning errors).

- Suggest adding 1–2 sentences linking survival analysis’ theoretical strengths， handling censored data, hazard modeling, or time-varying risk factors—to the nature of dialogue breakdowns in LLMs.

2. ***Problem Formulation***
- The definition of failure may be too coarse, as treating the first error as collapse ignores possible self-correction. Introducing a tolerance window or semantic distortion metric could yield a more continuous and realistic measure of failure.

3. ***Validation set:*** In the Experiment Setup section, the dataset is split into 80% for training and 20% for testing, but there is no validation set included.
A validation set is typically necessary for hyperparameter tuning and early stopping, so the authors should clarify whether they used cross-validation, a held-out subset of the training data, or another validation strategy to avoid potential overfitting.

4. The used benchmark in this paper only contains 700 questions, which is relatively small and thus lacks sufficient persuasive power to support the generality of the conclusions.

**Questions:**

1. ***Typos or inconsistency.*** The paper’s readability is poor, mainly due to weak continuity and unclear writing between sections. The transitions are abrupt, and ideas are not logically connected, making it difficult for readers to follow the narrative flow.

- Line 136 Page 3, "time-to-failure" should be "time-to-event".
- What exactly does the term **“feature”** refer to in this context in Table 1?
- The paper claims to analyze **9 state-of-the-art LLMs**, but based on Tables 1–2, it appears that **9 survival models were applied to a single LLM** instead. Could the authors clarify whether the study involved multiple LLMs or multiple survival modeling approaches on one model?

---

> ### Author Response · Authors · 2025-11-23
>
> We sincerely appreciate your time, patience, and high-quality constructive feedback. We have tried to address each of your specific points directly in the revision.
>
> For your convenience, here's our response to your concerns/questions:
>
> > W1: Missing Conceptual Depth
>
> We’ve rewritten the abstract and the beginning of Section 1 to make this explicit:
>
> -  clearly define the event as “first inconsistent answer” and time as the turn index, and explain that conversations can either end in success or failure → a classic time-to-event setting.
>
> - In the new paragraph added (From static accuracy to time-to-event in Sec.1), We highlight what survival analysis gives us that a plain classifier/regressor does not:
>   -  principled treatment of right-censored conversations (those that stay correct within the horizon),
>   -  hazard functions that show how failure risk changes as the dialogue progresses, and
>   -  time-varying covariates so we can connect evolving semantic drift to changing risk.
>
> We hope these changes directly address the requested conceptual bridge between conversational failure and survival analysis.
>
> >W2: Problem Formulation: The definition of failure may be too coarse, as treating the first error as collapse ignores possible self-correction. Introducing a tolerance window or semantic distortion metric could yield a more continuous and realistic measure of failure.
>
> We agree that there are richer ways to define conversational failure, but in this paper we intentionally adopted the MT-Consistency criterion **on purpose**—first deviation from the initial correct answer—for two reasons.
> - First, from a robustness and safety perspective (especially in domains like healthcare and education), whether a harmful or incorrect answer ever appears is often more critical than whether the model later self-corrects: once a wrong dosage, diagnosis, or fact is given, the downstream consequence or loss of trust may already have occurred. In that sense, the first inconsistent answer is a meaningful and conservative endpoint.
> - Second, this “single, well-defined event” aligns very naturally with a time-to-event formulation and keeps the survival process
> clean and interpretable. That said, we now explicitly acknowledge in the revised Discussion that this is a strict definition: it does not distinguish error types, nor does it model self-correction. We see these extensions as future work, including tolerance
> windows (e.g., allowing transient errors within a fixed span), graded semantic distortion scores instead of a binary event, and separate modeling of “time-to-self-correction” as a secondary event.
>
> > W3: Validation set: In the Experiment Setup section, the dataset is split into 80% for training and 20% for testing, but there is no validation set included. A validation set is typically necessary for hyperparameter tuning and early stopping, so the authors should clarify whether they used cross-validation, a held-out subset of the training data, or another validation strategy to avoid potential overfitting.
>
> Thank you for pointing this out— We did conduct the data split as well as the hyperparameter tuning in the experiments. To make it clear, we've updated the section (Section 4.3 Experiment Setup) and now clarify that we first create an 80% training pool and a 20% held-out test set, stratified by model and subject cluster. Within the 80% training pool, we then run 5-fold cross-validation over conversations to select model variants and any hyperparameters:
>
> For the advanced Cox/AFT models, the only hyperparameter is the strength of the $l_2$ penalty on the drift–model interaction terms. Both are chosen by 5-fold CV using IBS (with C-index as a secondary tie-breaker).
>
> For the Random Survival Forest, we tune the number of trees, maximum depth, and \texttt{mtry} using the same 5-fold CV procedure. All hyperparameter candidates and the best performance ones are now listed in Appendix E.
>
> >W4: The used benchmark in this paper only contains 700 questions, which is relatively small and thus lacks sufficient persuasive power to support the generality of the conclusions.
>
> It is true that MT-Consistency contains 700 base questions, but each is paired with up to 8 adversarial turns and evaluated across 9 LLMs, yielding **36,951 turns** QAs, which is still an effect size for analysis. We have clarified in Section 4.1 that:
> - The resulting dataset comprises thousands of conversations after filtering for initially correct answers (not just 700), and
> - The diversity comes from multiple subjects, difficulty levels, adversarial patterns, and model families.
>
> That said, we agree that our conclusions are tied to this particular benchmark and adversarial protocol. In the Limitations, we now explicitly state:
> - that all results are based on MT-Consistency with an 8-turn adversarial script, and
> - that extending the analysis to other benchmarks and longer, mixed-initiative conversations is an important direction for validating generality.

---

> ### Author Response · Authors · 2025-11-23
>
> > Q1-1: The paper’s readability is poor, mainly due to weak continuity and unclear writing between sections.
>
> We took this feedback seriously and made several structural changes:
> - The Introduction now ends with a concise list of three main contributions (survival framing, drift-aware dynamics, and AFT-based monitoring).
> - Section titles are reframed to answer clear questions (e.g., problem formulation, feature engineering, modeling framework, experiments, results, monitoring), and we added short “signposting” sentences at the start of each section explaining how it connects to the previous one.
> - We significantly tightened and clarified Section 3 (Methods), splitting it into: problem formulation, feature engineering, and survival modeling, each with explicit notation and definitions.
> - We moved some technical details (e.g., detailed AFT distribution formulas, PH tests) to the appendix and keep the main text flow focused on key ideas and findings.
>
> These edits are aimed directly at improving continuity and readability!
>
> >Q1-2: Typos and terminology (“time-to-failure”, “features”)
> We’ve replaced informal “time-to-failure” with the more standard “time-to-event” in the formal exposition. “Failure” is now used only as the informal label for that event.
>
> Col name in Table 1: “Features” was meant to be “number of covariates.”. We've updated it in the Table.1.
>
> >On the “9 LLMs vs 9 models” confusion
>
> We apologize for the confusion. The setup is:
> - Nine LLMs (Claude 3.5 Sonnet, DeepSeek R1, GPT-4o, an open-weight 120B GPT-style model (gpt\_oss\_120B), Llama 3.3 70B, Llama 4 Maverick, Gemini 2.5, Mistral Large, and Qwen 3.) are evaluated on the same adversarial conversations.
> - We pool conversations across all 9 models into one survival dataset and include model identity as a categorical covariate (with model–drift interaction terms in the advanced Cox and AFT variants).
> - Separately, we compare nine survival model specifications (two Cox variants, six AFT variants, one RSF) on this pooled dataset.
>
> We have clarified the LLM model list in Section 4.1 (Data - models) to avoid confusion. In previous draft, we mention the model type as $M_i$ at the beginning of the Sec.3.3. To make it clear, we are now explicitly state it at Sec. 3.2 - Additional covariates:  $M_i$ stand for LLMs.
>
>
> **We appreciate your thoughtful feedback in strengthening our work. We hope the carefully revised manuscript and responses address your concerns sufficiently, and we respectfully ask that you consider raising your rating in light of these improvements.**

---

> ### Author Response · Authors · 2025-11-26
>
> Dear Reviewer t9mq,
>
> We sincerely appreciate your time, patience, and constructive feedback. For your convenience, here is a concise summary of how we addressed each concern in our revision:
>
> **W1 (Conceptual Depth):** We really appreciate this insightful suggestion! We rewrote the abstract and early Section 1 to explicitly frame our setup as a time-to-event problem: the event is the first inconsistent answer and time is the turn index. We highlight what survival analysis contributes beyond standard prediction (principled handling of right-censoring, hazard functions over turns, and time-varying covariates linking semantic drift to changing risk).
>
> **W2 (Coarse Failure Definition):** We deliberately adopt MT-Consistency’s “first deviation from the initial correct answer” as a conservative endpoint, motivated by safety/robustness scenarios where any harmful/incorrect answer can be critical, such as healthcare or education domains.
>
> **W3 (Missing Validation Set):** We clarified our procedure in both rebuttal content and the revised version: 80/20 stratified train-test split, then 5-fold CV within the training pool for hyperparameter tuning. All hyperparameter candidates and selected values are now in Appendix E.
>
> **W4 (Small Benchmark):** We clarified in Section 4.1 that although MT-Consistency has 700 base questions, combining up to 8 adversarial turns and 9 LLMs yields 36,951 turn-level QAs after filtering, giving sufficient effect size for our analyses. We now make explicit in the Limitations that our conclusions are tied to this benchmark and adversarial protocol and that extending to other datasets and longer, mixed-initiative conversations is important future work.
>
> **Q1-1 (Readability):** We restructured the paper to improve continuity: the Introduction now ends with three concise contributions; section titles are framed as questions, with short signposting paragraphs; Section 3 is split into problem formulation, feature engineering, and survival modeling; and some technical details are moved to the appendix to keep the main narrative focused.
>
> **Q1-2 (Terminology)**: We replaced “time-to-failure” with “time-to-event” in the formal exposition, and corrected “Features” in Table 1 to “Number of covariates.” We also clarify that: (i) we evaluate nine LLMs on the same adversarial conversations and encode model identity (and its interaction with drift) as covariates, and (ii) we compare nine survival model specifications (two Cox, six AFT, one RSF) on MT-Consistency.
>
> We have carefully revised the manuscript to address all points you mentioned. We hope these improvements sufficiently resolve your concerns, and we respectfully ask that you **consider re-evaluating the score in light of these changes**.  And we are always here if you have any other concerns. Thank you again for helping strengthen our work!

---

### Author Response · Authors · 2025-12-04
**Summary of Revisions, Strengths, and Clarifications (1/3)**

Dear Reviewers and Area Chair,

**We are deeply grateful for the reviewers' insightful and constructive feedback, and we appreciate the considerable time and effort devoted to evaluating our work through both initial reviews and subsequent discussions. We also thank the newly assigned Area Chair for graciously accepting this additional responsibility despite the compressed timeline.**

For your convenience, we have organized our response as follows: A summary of rebuttal interactions, an overview of our contributions, the principal strengths recognized by reviewers, and our clarifications along with new experimental results added during the rebuttal period.

***
**Summary of Rebuttal Interactions**

We respect the ICLR team's decision to revert scores and disable further discussion given the special circumstances. For the new AC's reference, we summarize the interactions that occurred before the reversion.

Original Scores (format: overall rating, confidence)
- Reviewer y2g9: $(2,2)$
- Reviewer t9mq: $(4,3)$
- Reviewer KPhd: $(6,2)$
- Reviewer YR9z: $(8,2)$

Timeline of Interactions
- Nov 22-23: We submitted rebuttals addressing reviewer concerns
- Nov 24: Reviewer y2g9 responded after reviewing our updated paper, other reviews, and rebuttals $\rightarrow$ **raised score from (2,2) to (4,3)**
- Nov 26: Reviewer YR9z responded, noting our rebuttals clarified key concerns and new experiments reinforced confidence in their initial assessment → **maintained score (8,2)**
- Nov 28: Scores reverted.

***
**Paper Summary**
We present the first large-scale survival analysis of conversational AI robustness, modeling LLM failure as a time-to-event process across 36,951 turns from 9 state-of-the-art LLMs on the MT-Consistency benchmark. Our framework combines Cox proportional hazards, Accelerated Failure Time (AFT), and Random Survival Forest models with semantic drift features, revealing that abrupt prompt-to-prompt drift sharply increases failure risk while cumulative drift is counterintuitively protective—suggesting adaptation in conversations that survive multiple shifts. We demonstrate that lightweight AFT models can function as turn-level risk monitors, flagging 76% of failing conversations before the first inconsistent answer while maintaining low false-positive rates.

**Our Contributions**
- **Framing**: We formalize time-to-inconsistency as a survival analysis problem, providing a temporally-aware view of conversational robustness beyond single-turn and static multiturn metrics.
- **Drift-aware dynamics**: We introduce simple semantic drift signals as time-varying covariates and show that abrupt prompt-to-prompt drift sharply increases hazard, whereas cumulative drift is unexpectedly protective, suggesting adaptation in conversations that survive multiple shifts.
- **Methodology and safeguards**: We find that AFT models with model-drift interactions offer the best discrimination and calibration, that key drift features violate proportional hazards assumptions, and that lightweight AFT-based monitors can estimate turn-by-turn risk, pointing toward practical real-time safeguards for multi-turn deployments.

***
**Strengths Highlighted by Reviewers**
- **Novel framing of multi-turn robustness via survival analysis**: Reframes multi-turn LLM robustness as a time-to-event (survival) problem rather than single-turn/static evaluation. (Reviewers t9mq, KPhd, y2g9, YR9z)
- **Semantic drift as interpretable time-varying features**: Models prompt-to-prompt, context-to-prompt, and cumulative semantic drift as time-varying covariates capturing conversational dynamics. (Reviewer KPhd)
- **Large-scale, rigorous experimental setup**: Runs a 36,951-turn study across 9 models with Cox, AFT, and RSF, revealing that proportional hazards often do not hold for key features. (Reviewers KPhd, YR9z)
- **Actionable, deployment-relevant guidance**: Shows lightweight AFT models can act as practical safeguards and support real-time risk monitoring and drift mitigation. (Reviewers KPhd, y2g9)
- **Clear, accessible, and well-situated presentation**: Provides a self-contained, detailed method description and strong related work, making the paper clear and accessible even to non-experts in survival analysis. (Reviewers y2g9, YR9z)

---

### Author Response · Authors · 2025-12-04
**Summary of Revisions, Strengths, and Clarifications (2/3)**

***
**Clarifications and Additional Results**
- **Conceptual framing of survival analysis for multi-turn robustness**: We rewrote the abstract and the start of Section 1 to explicitly define the event as “first inconsistent answer” and time as the turn index, and to explain why conversations that either end in success or failure form a classic time-to-event setting. We also added an explicit paragraph (“From static accuracy to time-to-event”) highlighting what survival analysis adds beyond a standard classifier/regressor: principled handling of right-censored conversations, hazard functions over turns, and time-varying covariates that link semantic drift to changing risk. (Reviewer t9mq, Reviewer y2g9)

- **Failure definition and scope of "failure types"**: We clarified that we intentionally adopt MT-Consistency’s "first deviation from the initial correct answer" as a conservative failure event, motivated by safety/trust (e.g., a single wrong dosage can already cause harm). We now explicitly discuss in the Discussion that this definition does not distinguish sycophancy vs. context confusion vs. knowledge gaps or self-correction, and we outline concrete future extensions (tolerance windows, graded semantic distortion scores, and separate “time-to-self-correction” modeling). (Reviewer t9mq, Reviewer KPhd)

- **Train/validation/test split and hyperparameter tuning**: We clarified the experimental setup: conversations are first split into an 80% training pool and 20% held-out test set, stratified by model and subject cluster. Within the 80% pool, we run 5-fold cross-validation over conversations to select hyperparameters:
  -  Cox/AFT: strength of the ℓ₂ penalty on drift–model interaction terms (chosen via CV using IBS, with C-index as a tie-breaker).
  -  RSF: number of trees, max depth, and mtry tuned with the same 5-fold CV.
Candidate grids and best settings are now listed in Appendix E. (Reviewer t9mq)

- **Dataset size, single benchmark, and 8-turn limit (generality & scaling)**: We clarified that although MT-Consistency has 700 base questions, after filtering to initially correct answers and expanding over 8 adversarial turns and 9 LLMs, the analysis uses 36,951 evaluated turns, with diversity across subjects, difficulty bands, adversarial patterns, and model families. At the same time, we now state explicitly in Data and Limitations that all results come from a single benchmark and an 8-turn adversarial script, and that this constrains generality. We frame our contribution as a first, controlled demonstration of the survival-analysis lens and mark extensions to other multi-turn benchmarks, longer mixed-initiative dialogues, and size-graded model families (for scaling-law-style analysis) as key next steps. (Reviewer t9mq, Reviewer KPhd, Reviewer YR9z)

- **Clarifying model set vs. survival models and "features" in Table 1**: We explicitly list the 9 LLMs evaluated and clarify that we pool conversations across all 9 into a single survival dataset, including model identity as a categorical covariate with model–drift interaction terms in both Cox and AFT variants. Separately, we compare 9 survival model specifications (two Cox variants, six AFT distributions, one RSF). Table 1 has been updated so that the “Features” column is now clearly labeled as “number of covariates,” resolving confusion about what “features” meant and how it differs from “number of models.” We also explicitly state that Cox and AFT use the same interaction structure; only the link/distribution differs. (Reviewer t9mq, Reviewer y2g9)

- **Drift feature definitions and use of embeddings**: We substantially clarified Section 3.2 (feature engineering). We now explicitly define:
  -  Prompt embeddings as the embedding of the user prompt at turn t.
  - Context embeddings as the embedding of the full dialogue (user prompt + LLM response) prefix up to turn t–1 .
  - P2P drift as the cosine distance between consecutive user prompts.
  - C2P drift as the distance between the context embedding up to t–1 and the current user prompt.
  - Cumulative drift as the sum of P2P drift over turns.
This makes explicit that drift features are defined with respect to user prompts, while context incorporates both user and LLM responses.
(Reviewer y2g9)

---

### Author Response · Authors · 2025-12-04
**Summary of Revisions, Strengths, and Clarifications (3/3)**

- **Presentation, continuity, and terminology fixes**: In response to readability concerns, we: (i) restructured the Introduction to end with three clear contributions (survival framing, drift-aware dynamics, and AFT-based monitoring), (ii) retitled sections to answer clear questions and added short “signposting” sentences connecting each section to the previous one, (iii) split Methods into problem formulation, feature engineering, and survival modeling with explicit notation, and (iv) moved some technical details to the appendix to keep the main narrative focused on the core ideas and findings. We also replaced informal “time-to-failure” with the standard “time-to-event” in the formal exposition. (Reviewer t9mq, Reviewer y2g9)

- **Stronger analysis of drift effects and robustness to PH violations / difficulty confounds**: We expanded the empirical analysis around drift features:
  - We now report that prompt-to-prompt drift has large effect sizes (hazard ratios up to ~4–5× for some models) even after controlling for subject domain, difficulty band, context-to-prompt drift, cumulative drift, and model identity.
  - We added additional views (AFT and RSF variable importance, risk-stratified survival curves) to show that the “abrupt drift is risky, cumulative drift is protective” pattern is consistent across model families.
  - To address the PH violation for P2P drift, we fitted a Log-Normal AFT model and compared risk profiles: P2P remains a strong risk factor (acceleration factors < 1, shortening time-to-event), while cumulative drift remains protective (acceleration factors > 1), confirming that these are not artifacts of a mis-specified PH model.
  - We clarified that question difficulty enters as its own covariate in the survival models; the strong P2P effect persists after conditioning on difficulty, addressing the concern that high P2P might merely reflect harder follow-up questions. (Reviewer KPhd, Reviewer y2g9)

- **Empirical validation of AFT-based real-time monitoring**: We added a new experiment and subsection on retrospective risk monitoring with AFT. Using a trained Weibull AFT model, we compute a conditional failure probability (CFP) at each turn and trigger an alert when CFP exceeds a threshold. On a held-out set of 140 conversations, this monitor:
  - Alerts in advance on 76% of failing conversations, with a median lead time of 2 turns (mean ≈ 2.3) before the first inconsistent answer.
  - Triggers on only 19% of safe (censored) conversations, with ~0.5 alerts per safe conversation.
  - Outperforms a “drift-threshold” baseline, which detects fewer failing dialogues (62%) and produces more noise on safe ones (~1.2 alerts per safe conversation and higher alert rate overall).
These results provide concrete empirical support that survival-based monitoring can serve as a practical early-warning safeguard.
(Reviewer YR9z, Reviewer y2g9)

- **Explicitly stating limitations and future extensions around failure mechanisms and scaling**: We clarified that, in this first study, we intentionally treat “first inconsistent answer under MT-Consistency scoring” as a single event, regardless of whether it arises from sycophancy, confusion, or knowledge gaps, to keep the survival process clean and aligned with the benchmark. We now more explicitly mark fine-grained failure analysis (by mechanism) and re-running the pipeline on larger reasoning-focused models / size-graded model families as important follow-up directions rather than claims of the current paper. (Reviewer KPhd, Reviewer YR9z)

---

### Meta-Review · Area_Chair_pK6B · 2025-12-23

**Summary:**

The paper proposes applying Survival Analysis to evaluate the robustness of Large Language Models (LLMs) in multi-turn adversarial conversations. By modeling failure as a "time-to-event" process rather than a binary outcome, the authors identify that abrupt semantic drift (Prompt-to-Prompt) sharply increases failure risk, while cumulative drift acts protectively (suggesting adaptation). Reviewers praised the novelty of the framework and the rigor of the analysis (37k turns across 9 models). Initial concerns focused on presentation clarity (confusion over feature definitions), the definition of failure (too coarse), the reliance on a single benchmark (MT-Consistency), and the lack of empirical evidence for the claimed real-time monitoring capability. The authors provided a substantial rebuttal: they restructured the paper for clarity, explicitly defined features, conducted a new experiment demonstrating retrospective risk monitoring (using AFT models to predict failure with 76% sensitivity), and clarified statistical assumptions (Log-Normal AFT checks).

**Reviewer Concerns:**

**Addressed:**

1. Presentation & Feature Clarity (Reviewers y2g9, t9mq): The authors overhauled Section 3 and the Introduction to clarify that drift features are based on user prompts and to better motivate the "time-to-event" framing. Reviewer y2g9 explicitly acknowledged these improvements.

2. Real-time Monitoring Evidence (Reviewer YR9z, y2g9): The authors added a new experiment (Section 5.5) showing that their AFT model could act as a risk monitor, flagging 76% of failing conversations early. This directly addressed the demand for practical utility.

3. Drift vs. Difficulty (Reviewer KPhd): The authors clarified that "difficulty" is a controlled covariate, proving that the risk from abrupt drift is not just a proxy for harder questions.

4. Statistical Validity (Reviewer y2g9): The authors performed robustness checks using Log-Normal AFT models to confirm that the "protective cumulative drift" finding holds even when Proportional Hazards assumptions are violated.

**Outstanding:**

1. Single Benchmark Limitation (Reviewers t9mq, KPhd, y2g9): While the authors argued that the dataset is large (37k turns), the analysis is still restricted to the adversarial protocols of MT-Consistency. This limits the proven generality of the findings to other types of conversational degradation.

2. Causal Validation (Reviewer y2g9): The reviewer noted that while the correlation is strong, the paper lacks a causal intervention study (e.g., manipulating drift to cause failure) to fully validate the mechanism.

**Reviewer Scores:**

Reviewer YR9z: 8 (Score maintained)

Reviewer y2g9: 2 -> 4 (Score raised)

Reviewer KPhd: 6 (Likely maintained)

Reviewer t9mq: 4 (Likely maintained)

---

### Decision · Program_Chairs · 2026-01-26

Accept (Poster)